# The Multi-Ethnic New Zealand Study of Acute Coronary Syndromes (MENZACS): Design and Methodology

**Malcolm. E. Legget [1,2,\*], Vicky. A. Cameron [3], Katrina. K. Poppe [1,4], Sara Aish [1], Nikki Earle [1], Yeunhyang Choi [1,4], Kathryn. E. Bradbury [5], Clare Wall [6], Ralph Stewart [2], Andrew Kerr [4,7], Wil Harrison [7], Gerry Devlin [8], Richard Troughton [3], A. Mark Richards [3], Graeme Porter [9], Patrick Gladding [10], Anna Rolleston [1,11] and Robert N. Doughty [1,2]**

[1] Heart Health Research Group, Faculty of Medical and Health Sciences, University of Auckland, Auckland 1023, New Zealand; k.poppe@auckland.ac.nz (K.K.P.); s.aish@auckland.ac.nz (S.A.); n.earle@auckland.ac.nz (N.E.); yeunhyang.choi@auckland.ac.nz (Y.C.); anna@thecentreforhealth.co.nz (A.R.); r.doughty@auckland.ac.nz (R.N.D.)

[2] Greenlane Cardiovascular Service, Auckland City Hospital, Auckland 1023, New Zealand; r.stewart@adhb.govt.nz

[3] Christchurch Heart Institute, University of Otago, Christchurch 8011, New Zealand; vicky.cameron@otago.ac.nz (V.A.C.); richard.troughton@cdhb.health.nz (R.T.); mark.richards@cdhb.health.nz (A.M.R.)

[4] Epidemiology and Biostatistics, School of Population Health, Faculty of Medical and Health Sciences, University of Auckland, Auckland 1023, New Zealand; andrew.kerr@middlemore.co.nz

[5] National Institute for Health Innovation, School of Population Health, University of Auckland, Auckland 1023, New Zealand; k.bradbury@auckland.ac.nz

[6] Discipline of Nutrition, Faculty of Medical and Health Sciences, University of Auckland, Auckland 1023, New Zealand; c.wall@auckland.ac.nz

[7] Middlemore Hospital, Counties Manukau District Health Board, Auckland 2025, New Zealand; wil.harrison@middlemore.co.nz

[8] Gisborne Hospital, Tairawhiti District Health Board, Gisborne 4010, New Zealand; g.devlin@heartfoundation.org.nz

[9] Tauranga Hospital, Bay of Plenty District Health Board, Tauranga 3112, New Zealand; g.porter@bopdhb.govt.nz

[10] North Shore Hospital, Waitemata District Health Board, Auckland 0620, New Zealand; patrick.gladding@waitematadhb.govt.nz

[11] Centre for Health, Tauranga 3112, New Zealand

[\*] Correspondence: malcolm.legget@auckland.ac.nz

**Abstract: Background**. Each year, approximately 5000 New Zealanders are admitted to hospital with first-time acute coronary syndrome (ACS). *The Multi-Ethnic New Zealand Study of Acute Coronary Syndromes* (MENZACS) is a prospective longitudinal cohort study embedded within the All New Zealand Acute Coronary Syndrome Quality Improvement (ANZACS-QI) registry in six hospitals. The objective of MENZACS is to examine the relationship between clinical, genomic, and cardiometabolic markers in relation to presentation and outcomes post-ACS. **Methods**. Patients with first-time ACS are enrolled and study-specific research data is collected alongside the ANZACS-QI registry. The research blood samples are stored for future genetic/biomarker assays. Dietary information is collected with a food frequency questionnaire and information about physical activity, smoking, and stress is also collected via questionnaire. Detailed family history, ancestry, and ethnicity data are recorded on all participants. **Results**. During the period between 2015 and 2019, there were 2015 patients enrolled. The mean age was 61 years, with 60% of patients aged <65 years and 21% were female. Ethnicity and cardiovascular (CV) risk factor distribution was similar to ANZACS-QI: 13% Māori, 5% Pacific, 5% Indian, and 74% NZ European. In terms of CV risk factors, 56% were ex-/current smokers, 42% had hypertension, and 19% had diabetes. ACS subtype was ST elevation myocardial infarction (STEMI) in 41%, non-ST elevation myocardial infarction (NSTEM) in 54%, and unstable angina in 5%. Ninety-nine percent of MENZACS participants underwent coronary angiography and 90% had revascularization; there were high rates of prescription of secondary prevention medications upon discharge from hospital. **Conclusion**. MENZACS represents a cohort with optimal contemporary management and will be a significant epidemiological bioresource for

the study of environmental and genetic factors contributing to ACS in New Zealand's multi-ethnic environment. The study will utilise clinical, nutritional, lifestyle, genomic, and biomarker analyses to explore factors influencing the progression of coronary disease and develop risk prediction models for health outcomes.

**Keywords:** MENZACS; acute coronary syndrome; multi-ethnic; genomics; study design

## 1. Introduction

In New Zealand, 15% of all deaths annually are caused by ischaemic heart disease [1] and one in four major coronary events are fatal [2]. Although age standardised mortality rates for ischaemic heart disease have fallen dramatically since the late 1960s, New Zealand's mortality rates are still higher than many other western countries with persistent disparities based on ethnic group and social deprivation. Māori (the indigenous population of New Zealand) and Pacific peoples typically present with disease at a younger age, have higher readmission rates, and incur approximately double the European age-standardised mortality rate [3,4].

Whilst there have been considerable advances in the management of acute coronary syndromes (ACS), there are important knowledge and practice gaps in optimal risk stratification, treatment, and long-term outcomes for patients with cardiovascular disease (CVD) in New Zealand [5–7]. Recurrent event rates remain high, with a recent follow-up analysis of patients admitted with a first-time ACS showing that 15% experienced a non-fatal cardiovascular readmission and 16% had died within a year [8]. Identification of individuals or groups at high residual risk of further events, despite contemporary therapies, could lead to more targeted strategies to improve inequitable clinical outcomes. For patients admitted with a first-time ACS in New Zealand, there is a high incidence of premature disease with 25% being aged less than 55 years [4] and a very high burden of risk factors: half are current smokers, half have a BMI > 30 kg/m$^2$, and 16% have diabetes. Along with established clinical risk factors [9], it is becoming widely accepted that CVD risk prediction is improved by incorporating environmental and sociodemographic variables and their interactions with genetic and other omics markers [10].

However, genetic risk markers identified in international genome-wide association studies (GWAS) for risk assessment may not be ideal for translation to New Zealand's multi-ethnic population, since it is estimated that approximately 80% of all participants in GWAS are of European ancestry despite this group representing only 16% of the global population [11]. Population profiles of GWAS for coronary artery disease follow a similar pattern and are also primarily of European descent [11]. The transferability of this knowledge to other populations is now known to be problematic since populations vary in terms of allele frequency, effect size of risk variants [12,13], and having unique ethnic-specific genetic variants associated with disease risk. Moreover, genetic variants influence how different populations metabolise drugs [14–16] and this leads to disparate outcomes between ethnic groups, even when under the same treatment regimes.

The primary aim of the Multi Ethnic New Zealand study of Acute Coronary Syndromes (MENZACS) is to define the extent to which environmental and genetic factors contribute to the overall burden of ACS in New Zealand's ethnically diverse population presenting with first-time ACS. The study will utilise clinical, nutritional, lifestyle, genomic, and biomarker analyses to explore aetiological factors and to develop risk prediction models for outcomes.

The broad themes of the proposed research are as follows: (1) to explore the role of genetic variation in the progression of coronary disease in a contemporary cohort of New Zealanders with ACS; (2) to refine screening strategies in certain high-risk populations to enhance secondary risk prediction and early intervention using a combination of clinical risk factors, genomics, and biomarkers; and (3) to define how the response to therapies

for ACS differs by ethnicity across New Zealand's diverse ethnic population groups. The protocol and structure of the study are reported along with a description of the initial cohort.

## 2. Methods

### 2.1. Study Design and Participants

MENZACS was established in 2015 in New Zealand by the Heart Health Research Group, University of Auckland, in collaboration with the Christchurch Heart Institute, University of Otago. The study received national ethics approval in April 2015 from the Health and Disability Ethics Committee (Ref: 15/NTB/59), with each participant providing written informed consent, and the protocol is registered at the Australian New Zealand Clinical Trials Registry (ACTRN12615000676516).

MENZACS is a prospective longitudinal cohort study linked to the All New Zealand Acute Coronary Syndrome Quality Improvement (ANZACS-QI) electronic registry by using a common web-based platform, which records detailed clinical information and routine laboratory test results on over 95% of patients admitted with suspected ACS undergoing coronary angiography across hospitals in New Zealand [17]. The primary aim of ANZACS-QI is to support evidence-based management of ACS regardless of age, sex, ethnicity, socioeconomic status, or geographical domicile [18]. Anonymised linkage of registry data with national routinely collected data on hospitalisations, mortality, medication dispensing, and other administrative health data will enable extensive phenotyping of this patient cohort (Figure 1).

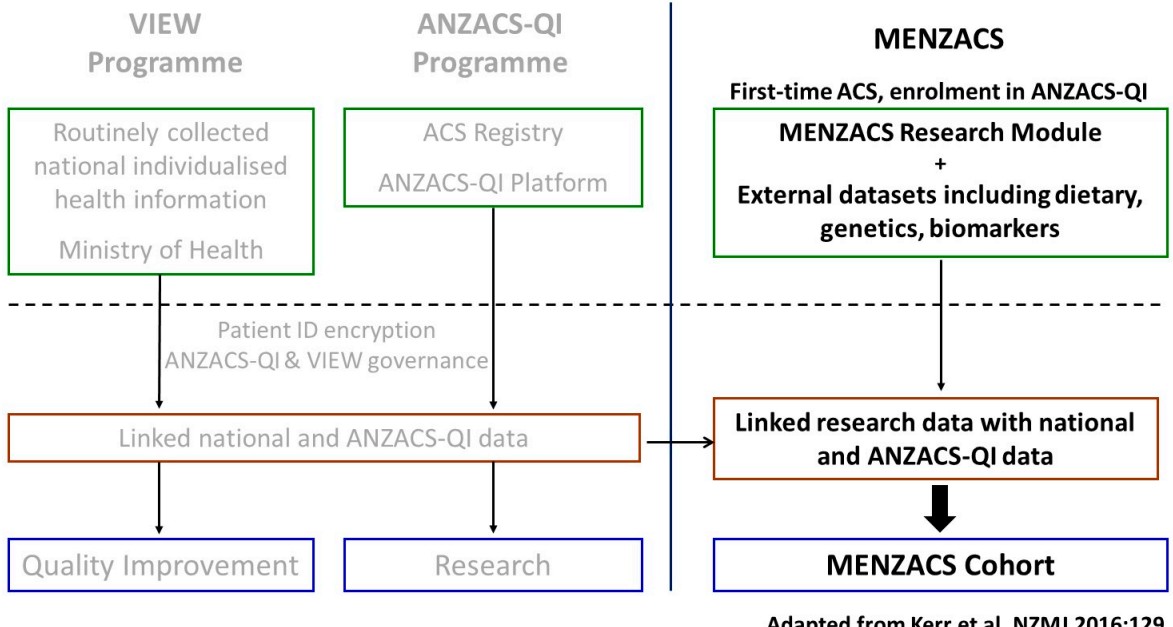

**Figure 1.** Schema showing the data platforms and linkages that result in the MENZACS cohort.

### 2.2. Study Infrastructure

A Māori Governance Group (MGG) was established at the beginning of this study. The MGG interact regularly with the research team and are kaitiaki (custodians) for Māori participants by providing advice and guidance using the principles of *Te Mata Ira—Cultural Guidelines for Biobanking & Genomic Research* [19] to ensure best practice. The patient information sheet and consent forms have been translated into Te Reo Māori, Tongan, and Samoan to help increase recruitment and engagement with Māori and Pacific patients. In addition, a kaupapa Māori information sheet is also available that more fully describes the cultural processes involved in the collection, storage, transportation, and destruction of samples.

### 2.3. MENZACS Inclusion Criteria and Recruitment Pathway

Patients aged >18 years with a clinical diagnosis of ACS are identified during the index hospital admission. Those fulfilling the criteria for a Type 1 myocardial infarction [20] or unstable angina [21] are eligible for enrolment. Patients are excluded if there is an elevation of troponin and/or if ECG changes are not thought to be due to an ACS, the patient has end stage renal failure (eGFR < 15 mL/min/m$^2$ or is receiving or planned to receive renal replacement therapy), is unable to give informed consent, or is not a New Zealand resident.

Patients admitted to the coronary care units of Auckland City, Middlemore, Christchurch, Waikato, North Shore, and Tauranga Hospitals are screened for eligibility into the study by a research nurse according to the criteria listed above. Eligible patients are invited to participate and are given the appropriate participant information and consent forms.

### 2.4. MENZACS Research Data

The routine clinically available data relating to the acute admission is extracted from the ANZACS-QI registry dataset and completed on all patients as part of the clinical workflow. These data include demographics, primary diagnosis, key cardiovascular risk factors, past cardiovascular history and admission vital signs, blood results, ECG, echocardiogram, in-hospital management, and coronary angiogram findings. MENZACS research-specific data are captured using a purpose-built module linked to the web-based ANZACS-QI platform, allowing efficient and standardised data collection at all participating sites. Data on physical activity was collated using the World Health Organisation Global Physical Activity Questionnaire [22] and information on stress and mood was gathered from a previously validated questionnaire of patients with stable coronary disease [23]. Other risk factors and clinical variables including smoking, history of gout, marijuana use, admission medication, and waist circumference were also captured.

Research and routine clinical data obtained by using the platform are linked to externally analysed research data on biomarkers, lipidomics, genetics, epigenetics, and diet. Specialised centres for analysis include the Christchurch Heart Institute-NT-proBNP and GDF-15, AgResearch (Invermay, Dunedin)-genotyping and the epigenetics analysis, AgResearch (Lincoln)-lipidomic analysis, and the Liggins Institute (University of Auckland)-Lp(a) concentrations. The results of these sample analyses are then sent to the MENZACS central data science management group at the University of Auckland for linkage to the main dataset accompanied with appropriate encryption processes. These data are then linked to national routinely collected data on past and future hospitalisations, dispensed medications, and death provided by the national Ministry of Health and curated by the *Vascular Information and the Web* (VIEW) programme of cardiovascular research (University of Auckland). Individualised patient data linkage was enabled by matching each patient's unique National Health Index (NHI) number to an encrypted NHI using a well-established protocol of de-identification, data security, and management that is central to the ongoing processes of the VIEW and ANZACS-QI programmes [18,24].This process allows long term follow up of 100% of the cohort that remained as residents in New Zealand.

Given the multi-ethnic emphasis of the study, in-depth data about family history, ancestry, ethnic background, iwi (tribal) affiliation, participants' and parents' country of birth, and grandparents' ethnicities are obtained. A dietary food frequency questionnaire (FFQ), with the wording of the food groups based on that used in the EPIC-Heart study [25] and adapted to the modern multi-ethnic New Zealand environment, was administered to all participants to survey dietary habits. This is a contemporary and culturally-appropriate questionnaire, which has been validated and shown to be reproducible in New Zealand adults [26], and is designed to enable comparison between cohorts and investigate the relationship between diet and chronic disease.

### 2.5. Biological Samples

A total of 40 mL of blood for genomic and biomarker analysis is drawn from each participant in a non-fasting state and seated. The time of acquisition of this blood sample

in relation to the time of admission was recorded. The samples are centrifuged (at 4 °C, 4000 rpm for 10 min) and separated as required into cryogenic tubes and frozen within 30 min of collection. The resulting EDTA and lithium heparin plasma, serum, and whole blood samples are stored at −80 °C in a secure dedicated facility in the University of Auckland School of Medical Sciences under the aegis of Te Ira Kāwai Auckland Regional Biobank framework (https://www.aucklandregionaltissuebank.ac.nz/ accessed on 19 April 2021) or at the Christchurch Heart Institute's Health and Disability Ethics Committee (HDEC) accredited tissue bank. All samples are logged in a specimen tracking and storage system. This is via OpenSpecimen (Krishagni Solutions Pvt Ltd., Pune, India) in Auckland or STARLims (Abbott Informatics, Hollywood, FL, USA) in Christchurch.

For Māori, the separation of body parts, tissues, or fluids from the person is acknowledged as an important cultural consideration and appropriate protocols are involved. In terms of using tissue or fluid samples for research it is accepted that those samples and the DNA extracted from them are taonga (gifts) and are therefore tapu (sacred). A kaupapa Māori research protocol information sheet has been created by the Māori Governance Group which outlines the appropriate tikanga (Māori custom) processes developed to ensure that the blood samples are treated with care and respect. There are processes around karakia (blessing) that is usually performed prior to a sample being destroyed or sent away for analysis.

### 2.6. Biochemical and Genetic Analyses

Laboratory results from tests undertaken as part of routine clinical care are available to the study and this includes creatinine, high sensitivity troponin, and lipid profiles. These have been recorded for >98% of the cohort. HbA1c is assayed when clinically indicated and has been recorded in 79% of the cohort. Additional research assays will measure other established and emerging risk markers and will include N-terminal pro B-type natriuretic peptide (NT-proBNP), Growth Differentiation Factor 15 (GDF-15), lipoprotein(a), and untargeted lipidomics using a mass spectrometry based lipidomics platform. This will measure more than 300 lipid molecules in plasma, which includes sphingolipids, phospholipids, glycerolipids, ceramides and di- and triglycerolipids, and cholesterol esters.

Genomic DNA on all participants has been extracted from 1 mL frozen whole blood using an automated QiaSympony DNA extraction process. Genome-wide genotyping will be performed using the Illumina Infinium Global Screening Array (640k SNPs, Illumina Inc., San Diego, CA, USA). This platform has been selected based on its ability to process samples within New Zealand and has high imputation accuracy at minor allele frequencies of >1% across multiple populations and includes curated clinical research variants and quality control markers. Epigenetic analysis will provide DNA methylation profiles in a subset of 1015 MENZACS participants (including all Māori participants) and will be performed using the Illumina Infinium Human Methylation EPIC beadchip array.

### 2.7. Clinical Outcome Variables

Clinical outcomes are defined from the national registries of ICD-10-AM (International Statistical Classification of Diseases and Related Health Problems, Tenth Revision, Australian Modification) coded hospitalisations and death and ACHI (Australian Classification of Health Interventions) coded procedures. A primary outcome of interest is major adverse cardiac events (MACE) defined as coronary revascularisation, readmission for cardiovascular cause including recurrent ACS, and death. Specific secondary outcomes include fatal or non-fatal ACS, fatal or non-fatal stroke or transient ischaemic attack (TIA), cardiovascular death, and all-cause death. Cardiovascular death will be defined from the ICD-10 coded death certificate or if death had occurred within 28 days of a CV hospitalisation.

### 2.8. Key Research Questions

This study will act as a resource for genomic discovery as well as providing a comprehensive study of the environmental, genetic, and conventional risk factors that are

associated with ACS in New Zealand. Whilst the data will allow "hypothesis free" unbiased discovery studies (see below), specific research questions and themes include:

- What dietary, lifestyle, and socio-economic factors are associated with first time ACS and subsequent outcomes;
- Can risk stratification be refined using clinical, biomarker, genetic, and epigenetic factors to build on existing secondary risk equations in New Zealand [27];
- Association studies of genetic variants with first time ACS and subsequent outcomes;
- The interaction of genomics, environmental factors, and biomarkers associated with certain phenotypes (e.g., diabetics, metabolic syndrome, hypertension, and obesity);
- Pharmacogenetic variability and the frequency of certain known variants across a diverse New Zealand population;
- Ethnic variation in genomic and genetic "signatures" related to cardiovascular risk factors.

### 2.9. Statistical Approach

A Data Science Advisory Group (DSAG) has been formed to provide specialist expertise in theoretical and applied statistics; this includes statistical genetics and will guide all analyses. Key analytical considerations include assessment of biological variation within and between data sources, reducing data dimensionality, time-to-event analyses, and the development of incremental risk scores for clinical use.

Existing clinical [28] and polygenic risk scores [29–33] will provide a starting point for the development of new incremental risk scores. In collaboration with data source specialists in nutrition, biomarkers, lipidomics, genetics, epigenetics, clinical science, and national health data repositories, the DSAG will discuss and advise on specialty-specific approaches to data reduction and analyses. The outputs of these analyses will inform how the data sources are best represented in the model, how model performance is assessed, and what model structure will be used. All data science will have input from the MGG on the appropriate use and interpretation of data in the Māori and Pacific context.

In the current report, key descriptors of the MENZACS study cohort (Table 1) have been presented as mean ± standard deviation, median (interquartile range), or frequency (percentage) as relevant. Any other analyses performed to this point have focused on the extent of missing data—for demographic and clinical variables and for the earlier quality assurance assessments of dataset linkage and the calculation of a published genetic risk score [31].

**Table 1.** Baseline demographics of the MENZACS cohort.

|  | *n* | Denominator [#] |
|---|---|---|
| Age, years | 61 (53, 69) | 2015 |
| Male | 1589 (78.9) | 2015 |
| Ethnicity |  | 2015 |
| Māori | 259 (12.8) |  |
| Pacific | 104 (5.2) |  |
| Indian | 94 (4.7) |  |
| Chinese | 13 (0.6) |  |
| European | 1499 (74.4) |  |
| Other * | 46 (2.3) |  |

**Table 1.** *Cont.*

|  | *n* | Denominator [#] |
|---|---|---|
| NZDep Index, quintile |  | 2015 |
| 1 (least deprived) | 479 (23.8) |  |
| 2 | 416 (20.6) |  |
| 3 | 361 (17.9) |  |
| 4 | 374 (18.6) |  |
| 5 (most deprived) | 385 (19.1) |  |
| Charlson comorbidity score |  | 2015 |
| 0 (no comorbidity) | 1801 (89.4) |  |
| 1–2 (moderate) | 189 (9.4) |  |
| ≥3 (severe) | 25 (1.2) |  |
| Diabetes on admission | 368 (18.5) | 1986 |
| COPD | 126 (6.4) | 1984 |
| Smoking status |  | 2014 |
| Never smoked | 838 (41.6) |  |
| Ex-smoker | 700 (34.8) |  |
| Current smoker | 476 (23.6) |  |
| BMI, kg/m$^2$ | 29.5 ± 5.6 | 2015 |
| ≥30 kg/m$^2$ | 788 (39.1) |  |
| TC: HDL | 4.7 ± 1.7 | 1872 |
| ≥4 | 1209 (64.6) |  |
| LDL cholesterol, mmol/L | 3.0 ± 1.4 | 1985 |
| eGFR, mL/min/1.73 m$^2$ | 79 (68, 92) | 1967 |
| HbA1c, mmol/mol |  | 1610 |
| Diabetes | 62 (50, 79) | 320 |
| No diabetes | 38 (35, 40) | 1290 |

Values are *n* (column percentage), median (interquartile range), or mean ± standard deviation. [#] Due to the nature of the registry, some variables were not mandatory and resulted in limited missing data which is represented by the denominator. NZDep = New Zealand Socioeconomic Deprivation; COPD = chronic obstructive pulmonary disease; BMI = body mass index; TC:HDL = ratio of total to high-density lipoprotein cholesterol; LDL = low-density lipoprotein; eGFR = estimated glomerular filtration rate, HbA1c = glycosylated haemoglobin. * "Other" ethnicities are non-Chinese Asian, Middle Eastern, Latin American, African, or Other.

## 3. Study Progress

The MENZACS study commenced with a pilot phase of recruitment at Auckland City Hospital (ACH) in July 2015. Middlemore and Christchurch started recruitment in March 2016, Waikato Hospital in May 2016, North Shore Hospital in May 2018, and Tauranga Hospital in July 2018. The first phase of recruitment was completed in July 2019. Of the 4846 patients who were screened and met the inclusion criteria, 229 were excluded and 2601 were not enrolled due to patient or logistic issues and one withdrew, leaving a total of 2015 patients included in this cohort (Figure 2). Fifty-seven patients did not have a diagnosis of confirmed ACS in the ANZACS-QI registry, and, thus, an adjudication process was undertaken by designated cardiologists at each centre who reviewed the clinical records and determined whether ACS occurred or not (including the type of ACS event). This resulted in 11 of these patients being excluded as confirmed non-ACS events.

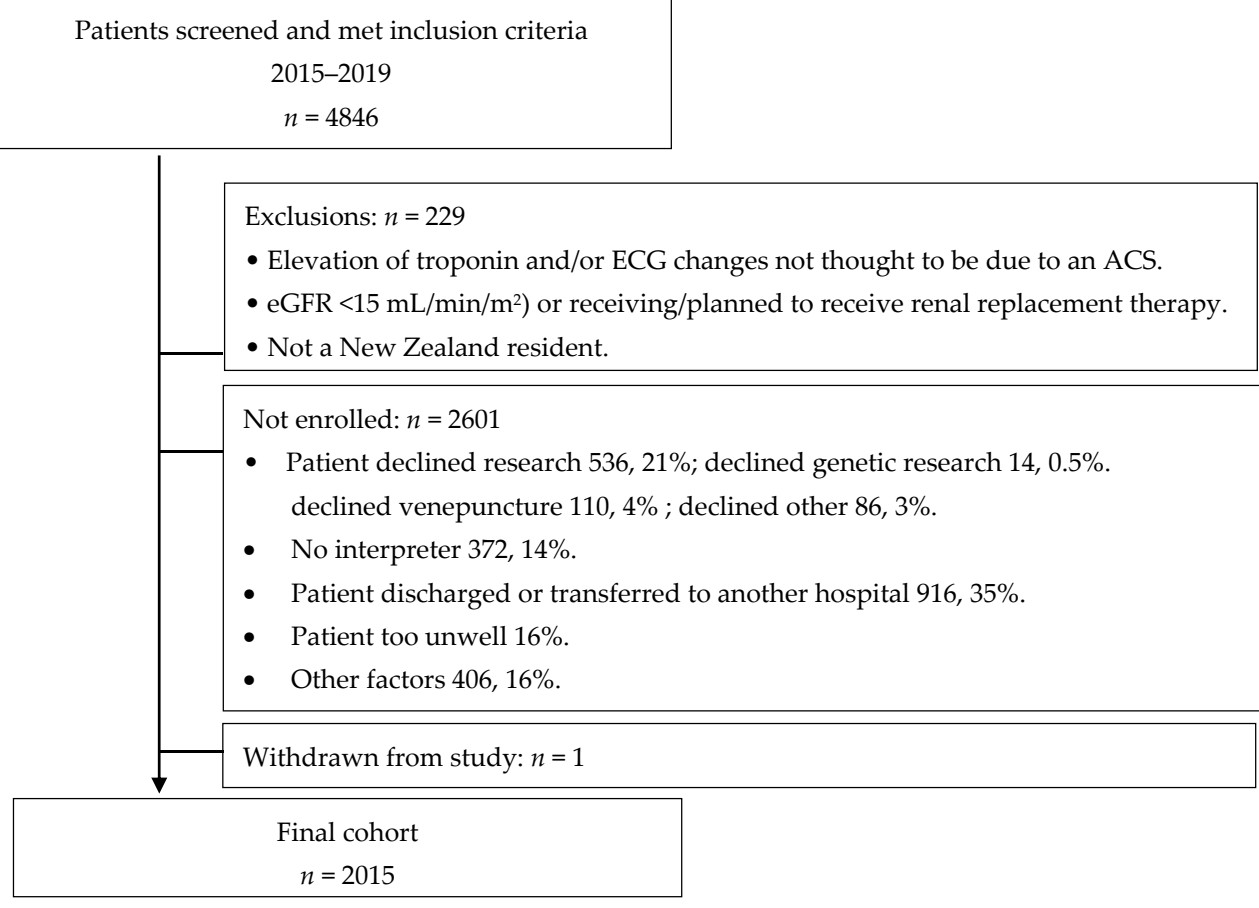

**Figure 2.** Flow diagram of screening and enrolment into the MENZACS.

### 3.1. Baseline Characteristics

The baseline demographics of the initial MENZACS cohort are shown in Table 1. The median age was 61 years, with 61% aged <65 years, and the cohort was predominantly male (79%). Māori constituted 13% of the cohort, Pacific and Indian peoples each comprised 5%, and 74% of participants were of European ethnicity. Based on the ethnic distribution and demographics of a contemporaneous cohort of first-time acute coronary syndrome patients in the ANZACS-QI registry, there is unlikely to have been an ethnic or risk factor bias in enrolment status in MENZACS [4].

Nineteen percent were in the most deprived quintile, 19% had diabetes, and 24% were current smokers. Almost all patients underwent angiography during the index admission and revascularisation rates were high, with 90% of the cohort undergoing either PCI or CABG during their hospital admission. These rates are consistent with the recruiting centres for MENZACS being secondary and tertiary referral centres for coronary intervention. Comprehensive information on extent of coronary artery disease, left ventricular function, GRACE score [34], and admission and discharge medication was obtained (Table 2).

### 3.2. Quality Assessment

#### 3.2.1. Data Linkage

A quality assurance assessment using data on the first 500 participants was performed. It confirmed that all MENZACS data sources can be linked via an encrypted NHI number. This involved transferring and merging the genetic database with MENZACS and ANZACS-QI data held at the National Institute of Health Innovation, University of Auckland (NIHI) and similarly, transferring, and merging diet data. A MENZACS study encrypted NHI was applied to the final merged dataset.

**Table 2.** Baseline clinical characteristics of the MENZACS cohort.

|  | *n* (%) | Denominator |
|---|---|---|
| Type of acute coronary syndrome |  | 2015 |
| STEMI | 827 (41.0) |  |
| Non-STEMI | 1083 (53.8) |  |
| Unstable angina | 105 (5.2) |  |
| Coronary angiography | 1968 (99.1) | 1985 |
| Ejection fraction assessed | 1824 (92.4) | 1974 |
| Ejection fraction <50% | 693 (35.1) |  |
| GRACE in-hospital score |  | 1984 |
| Low <1% | 580 (29.2) |  |
| Intermediate 1–2% | 864 (43.6) |  |
| High ≥3% | 540 (27.2) |  |
| Management |  | 1971 |
| PCI | 1408 (71.4) |  |
| CABG | 371 (18.8) |  |
| Medications on admission |  | 2015 |
| Statin | 493 (24.5) |  |
| Beta-blocker | 241 (12.0) |  |
| ACEi/ARB | 566 (28.1) |  |
| Aspirin | 324 (16.1) |  |
| Medications on discharge |  | 1972 |
| Statin | 1922 (97.5) |  |
| Beta-blocker | 1652 (83.8) |  |
| ACEi/ARB | 1550 (78.6) |  |
| Aspirin | 1927 (97.7) |  |
| Other antiplatelet agent | 1636 (83.0) |  |
| Dual antiplatelet therapy | 1611 (81.7) |  |

STEMI = ST-elevation myocardial infarction; GRACE = Global Registry of Acute Coronary Events; PCI = percutaneous coronary intervention; CABG = coronary artery bypass graft; ACEi = angiotensin-converting-enzyme inhibitor; ARB = angiotensin receptor blocker.

### 3.2.2. DNA Quality and Quantification

By using 100 samples from four sites, genomic DNA was isolated from 3 mL of frozen whole blood. All samples had DNA yields sufficient for genotyping (≥50 ng/μL) and 96% of samples had 260/280 ratios > 1.7, which indicated high purity. Genotyping was performed for 23 coronary-disease associated or gender-related single nucleotide polymorphisms (SNPs) using an Agena Bioscience MassARRAY®. A genetic risk score based on published methods [31] was able to be calculated for 97% of samples and gender was correctly identified in 100% of samples (see Supplementary Figure S1).

### 4. Discussion

MENZACS has been established as a significant epidemiological bioresource for the study of environmental and genetic factors contributing to ACS in New Zealand's contemporary ethnically diverse population. Genomic research into cardiovascular disease in New Zealand with the Post-Myocardial Infarction Study and Coronary Disease Cohort Study have already contributed valuable insights into genomic risk variants associated with clinical outcome and age of onset of CVD [35,36]. Building on this experience, the

MENZACS cohort will have comprehensive data to gain a better understanding of genetic influence on the varying phenotypic presentations of ACS in New Zealand's population and will explore the clinical utility of genomics in predicting secondary outcomes. The strength of this registry-based study is in the breadth and depth of data gathered, which will enable research into clinical, nutritional, genomic, lipidomic, and epigenetic factors influencing secondary outcomes.

The current MENZACS cohort reflects a population of patients who are treated intensively at the time of their index ACS admission, with 99% undergoing coronary angiography, 90% having coronary revascularization, and the vast majority prescribed statin (97%) and dual antiplatelet therapy (82%) at the time of hospital discharge. Compared to the ANZACS-QI registry cohort of patients with first-time ACS over a similar time period (January 2015 to December 2016), there were slightly more Māori participants and more patients had STEMI [8]. However, the overall demographic differences are small and reflect a referral population in the enrolment centres in the current study. Importantly, with the requirement to represent patients who undergo coronary angiography, ANZACS-QI only captures approximately 60% of New Zealanders admitted to hospital with their first ACS and there are important differences in patient characteristics and outcomes between those who are and are not included in the registry. Patients who are not captured in ANZACS-QI are older, are more commonly women with a higher comorbidity burden, and are more than twice as likely to experience death or a non-fatal CV readmission within 12 months of the index ACS admission [8].

The initial phase of this research has highlighted several findings relevant to the study of acute coronary disease in New Zealand. Firstly, MENZACS has served as a "proof of concept" that research studies can be successfully embedded within a web-based electronic registry used for clinical quality improvement purposes. The level of data capture was very high with minimal missing data and successful linkage of multiple datasets; this enabled a cost effective and efficient means of running a registry-based research study in an acute care environment. Linkage to national and routinely collected health data has allowed anonymised long-term follow-up of patient outcome, rehospitalisation, and medication dispensing. One limitation of the study is that there can be no feedback of information to an individual patient due to the anonymisation process. However, participants can opt to be approached for future research studies as part of the consent process.

Another limitation of the study is that a relatively high percentage of screened patients were not able to be enrolled in the study. The logistic issues related to performing a study such as MENZACS in tertiary and secondary referral centres have been significant. Patients are often transferred back to the referring hospital soon after coronary intervention at regional centres and may be sedated or are too unwell to enroll in the study. In addition, the length of hospital stay in the contemporary era of ACS management is short, resulting in a limited time period to approach patients for research during an acute index ACS admission. Overall, the willingness to participate in a study involving donating DNA has been very high and the research team has worked hard to ensure appropriate and detailed explanation is given in a culturally appropriate context. The Māori Governance Group has been integral to guiding culturally appropriate study processes, including conceptualising the study goals, facilitating recruitment strategies, developing sample handling and disposal protocols, and data governance.

The vast majority of studies examining the genetic contribution to the risk of secondary events in those with established coronary artery disease have been in European populations, with the largest being the GENIUS-CHD consortium involving over 185,000 participants [37,38]. However, there are far fewer studies of non-European populations who are at particularly high risk [39,40]. The predictive value of polygenic risk scores derived from cohorts of predominantly European ancestry can be attenuated in other ethnic groups and this emphasises the need for well phenotyped studies involving indigenous populations [11,41,42]. This is a crucial goal as genomic research focussed on European populations can compound inequity when applying the results of the research, for example, in

the utility of risk prediction and health prevention strategies [41,43–45]. The high incidence of premature CVD and worse cardiovascular health outcomes among some ethnic groups and a single national public healthcare system with a unique patient identification number renders New Zealand ideally suited to study the genetic and environmental drivers and influences on cardiometabolic outcomes.

The first phase of MENZACS analyses will focus on the association of established cardiovascular risk factors with secondary cardiovascular outcomes, and the development of incremental risk prediction tools utilising genetic, biomarker, lipidomic, and epigenetic markers. Subsequent studies will examine inter-ethnic variation, nutritional, lifestyle, pharmacogenomic, and kaupapa Māori approaches to optimising outcomes following an ACS, and assist in personalisation of risk stratification, and therapeutic intervention.

**Supplementary Materials:** The following are available online at https://www.mdpi.com/article/10.3390/cardiogenetics11020010/s1, Figure S1: Distribution of 27 SNP genetic risk score by gender.

**Author Contributions:** Conceptualization, M.E.L., V.A.C., R.N.D., K.K.P., N.E. and A.R.; methodology, M.E.L., V.A.C., R.N.D., K.K.P., N.E. and A.R.; writing—original draft preparation M.E.L., V.A.C., R.N.D., K.K.P., N.E., A.R., review and editing, M.E.L., V.A.C., R.N.D., K.K.P., N.E., A.R., S.A., Y.C., K.E.B., C.W., R.S., A.K., W.H., G.D., R.T., A.M.R., G.P., P.G. All authors have read and agreed to the published version of the manuscript.

**Funding:** The MENZACS study is supported by grants from the Heart Foundation (Heart Health Research Trust grant 1957), Healthier Lives National Science Challenge (Ministry of Business Innovation and Employment Reference UOOX1902), Green Lane Research and Educational Fund (17/26/4130), Freemasons Foundation, and the University of Auckland. R.N.D. is the holder of the Heart Foundation Chair of Heart Health; K.K.P. is the holder of the Heart Foundation Hynds Senior Fellowship; N.E. holds a NZ Heart Foundation Post-doctoral Research Fellowship; K.E.B. holds a Sir Charles Hercus Health Research Fellowship from the Health Research Council of New Zealand.

**Institutional Review Board Statement:** The study was conducted according to the guidelines of the Declaration of Helsinki, and approved by the New Zealand Northern B Health and Disability Ethics Committee (Ref: 15/NTB/59) 14 April 2015.

**Informed Consent Statement:** Informed consent was obtained from all subjects involved in the study.

**Data Availability Statement:** Restrictions apply to the availability of these data.

**Acknowledgments:** MENZACS Executive Group; M Legget * (Chair and Co-PI), V Cameron (Co-PI), S Aish * (Project manager), R Doughty *, N Earle *, K Poppe *, A Rolleston, C Wall. * Coordinating Centre. MENZACS Steering Group; M Legget, R Doughty, R Stewart, A Kerr, W Harrison, G Devlin, V Cameron, R Troughton, AM Richards, S Aish, K Poppe, C Wall, G Porter, and P Gladding. International Advisors. J. Danesh (Cambridge, UK), and J Howson (Oxford, UK). MENZACS Māori Governance Group; A Rolleston (Chair), K Southey, K Henare, R Stewart, C Grey, and H Wihongi. MENZACS Data Science Group; K Poppe (Chair), N Earle, J Howson, T Lumley, and A Pilbrow. Study centres; University of Auckland (M Oakes-Ter Bals, M Heath, P Shepherd, A Rykers, T Frugier, J Copedo, B Wu, Y Jiang, B Seers, A Chaptynova, C Fyfe, S Wall, and N Kluger). Auckland City Hospital (R Stewart, and K Marshall). Christchurch Hospital and Christchurch Heart Institute (V Cameron, A Pilbrow, S Prue, L Skelton, and R Troughton,). Waikato Hospital (C Nunn, G Devlin, V Pera, L Lowe, S Pilkington, and G Francis). Middlemore Hospital (A Kerr, L Pearce, M Ma, R Railton, L Sharp, P Sharma, and J Gilmore). Tauranga Hospital (G Porter, J Goodson, J Shippey, J Tisch, K Presley, T McKenzie, and CCU Staff). North Shore Hospital: T Scott, G McAnnalley, C Hulbert, K Smith, C Campbell, K Stanley, C Clow, and J Chen. AgResearch: K. Fraser. Enigma Solutions Ltd.: S Breen, and C Wiltshire. The MENZACS investigators would like to express their deepest gratitude to all the patients who have participated in the study.

**Conflicts of Interest:** RND has received research grants (administered through host institution the University of Auckland) from the NZ Heart Foundation, Health Research Council of New Zealand, Roche Diagnostics and Bayer. AMR holds grants and/or aid in kind and has received speaker fees and Advisory Board fees from Roche Diagnostics, Abbot Labs, Sphingotec, Critical Diagnostics, and Thermo Fisher and has received biomarker study support in kind and/or as grants from AstraZeneca and Bristol Myers Squibb. RT has received grant funding and consulting fees from Roche Diagnostics.

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
