# Peer review of "The Multi-Ethnic New Zealand Study of Acute Coronary Syndromes (MENZACS): Design and Methodology"

_cardiogenetics, doi:10.3390/cardiogenetics11020010_

Round 1

Reviewer 1 Report

The authors of “The Multi-Ethnic New Zealand study of Acute Coronary Syndromes (MENZACS): Design and methodology” present a new cohort of first-time ACS patients in New Zealand to study the associations between clinical, genomic and other biomarkers with events in the follow-up. The article is interesting and well written. However, some aspects should be improved:

Methods

Authors said that physical activity, stress and mood, smoking, waist circumference, history of gout, marijuana use, and admission medication were obtained using a questionnaire similar to previous studies of coronary disease cohorts. Were the published questionnaires used directly or were they modified? If the questionnaires were modified, were they validated?

They say that data on biomarkers, lipidomics, genetics, epigenetics and diet are externally analyzed. Where are these data analyzed?

Authors describe that laboratory results from tests undertaken as part of routine clinical care are available to the study. Are they available by linkage to clinical records?

Statistical analysis does not describe the descriptive analysis presented in the manuscript.

Results

2,601 were not enrolled due to patient or logistic issues: this is a very high number, which were the issues? The participants not enrolled are from a particular ethnicity?

It would be interesting to examine demographic be patient subgroups, for example in patients who survived and in patients who died.

It seems that most participants are from European ethnicity (74%), while 13% and 5% are Maori and Pacific/Indian. Is this representative of the population of New Zealand?

Authors state that data validation and cross-sectional analyses of a dataset including genetic and diet information was performed: why this information is not presented? If it is described in the results section in should be shown.

Similarly, the authors comment that a genetic risk score based was able to be calculated for 97% of samples and gender was correctly identified in 100% of samples: this information should also be presented and risk score calculations should be described in the statistical analysis section.

Discussion

Authors compare the cohort characteristics with the ANZACS-QI registry cohort. It would be interesting to compare this new cohort with cohorts from other regions.

Authors say that ANZACS-QI captures approximately 60% of New Zealanders admitted to hospital with their first ACS. Could this be a problem for identifying follow-up events if this is the source of hospital events and mortality? This should be highlighted as a limitation.

Authors state that linkage to national routinely collected health data has allowed anonymized long-term follow-up of patient outcome, re-hospitalization, and medication dispensing. However, this information is not shown.

Tables

In Table 1, the second column should not have the heading n (%) as not all variables are categorical and described with frequencies.

Author Response

Please see the attachment- Response to Reviewer 1 Comments.docx

Reviewer 2 Report

In this manuscript Legget et al present the design and methodology of the Multi-Ethnic New Zealand study of Acute Coronary Syndromes (MENZACS). The manuscript is well written and may be of interest to readers of the journal. The authors should be complimented for their efforts in trying to clarify the role of clinical en genomic variables in relation to presentation and outcome after ACS. Especially in current times, where discussing ethnic differences is an (over)sensitive topic, the investigators manage to address this in a very respectful way, as they have also described extensively throughout their manuscript. I do however have some questions about the design of the study:

  1. First, the authors want to investigate the role of clinical en genomic variables in relation to presentation and outcome after ACS irrespective of socioeconomic status or geographical domicile. How do the authors plan to correct for all these variables? Perhaps these are presented elsewhere, but I think this is of crucial importance in this manuscript, especially with regard to outcome. What are the differences in early access to a primary PCI center? What is the effect of education level in recognizing complaints and the decision to seek medical help? As the time between start of complaints and opening of the culprit coronary artery is a very strong predictor of outcome, this need to be taken into account, especially in relation to geographical domicile (which in turn is also related to ethnic background. I am very interested on how the authors plan to correct for these kind of variables.
  2. The investigators lose around 54% of their patient inclusion due to "patient or logistic issues". I find this a very high percentage... Although this cannot be changed anymore, I think it should be mentioned as a limitation in the discussion section of the manuscript.

Author Response

Please see the attachment- Response to Reviewer 2.docx

Round 2

Reviewer 1 Report

First, I want to advise the authors to carefully check the files that they upload in the review processes. They uploaded a file full of comments to authors and with some sentences that are not finished or that do not make sense.

On the other hand, the authors correctly answered to most of the comments. However, there are some minor comments that need to be solved:

  • I assume that part of the information provided for points 1-2 will be added to the manuscript.
  • Point 4: The descriptive statistical analysis has not been provided.
  • Point 5: The authors explained the reasons for not enrollment. I was wondering if they did a descriptive analysis of the not enrolled patients to see if there was any difference in risk factors or other variables between enrolled and not enrolled.
  • The response to Point 9 is not clear.
  • Point 10: I still think that the authors should compare their cohort with cohorts from other regions in the discussion.

Reviewer 2 Report

My questions have been answered appropriately.

However, the reply to the other reviewer still contains internal discussion of the research group and seems to be an unfinished document. Please complete this and resubmit.
